# Cicadomorpha Community (Hemiptera: Auchenorrhyncha) in Portuguese Vineyards with Notes of Potential Vectors of *Xylella fastidiosa*

**DOI:** 10.3390/insects14030251

**Published:** 2023-03-02

**Authors:** Isabel Rodrigues, Maria Teresa Rebelo, Paula Baptista, José Alberto Pereira

**Affiliations:** 1Centro de Investigação de Montanha (CIMO), Instituto Politécnico de Bragança, Campus de Santa Apolónia, 5300-253 Bragança, Portugal; 2Laboratório Associado para a Sustentabilidade e Tecnologia em Regiões de Montanha (SusTEC), Instituto Politécnico de Bragança, Campus de Santa Apolónia, 5300-253 Bragança, Portugal; 3Departamento de Ingeniería Agrária, Universidad de Léon, Av. Portugal, n° 41, 24071 Léon, Spain; 4Centre for Environmental and Marine Studies (CESAM), Department of Animal Biology, University of Lisboa, Campo Grande 016, 1749-016 Lisboa, Portugal

**Keywords:** *Cicadella viridis*, *Philaenus spumarius*, Pierce’s disease, Cicadellidae, inter-row vegetation

## Abstract

**Simple Summary:**

Cicadomorpha (Hemiptera: Auchenorrhyncha) represent a group of concern in vineyards since they can cause damage directly through feeding or can be vectors for serious pathogens such as grapevine yellows or the bacterium *Xylella fastidiosa*, the causal agent of Pierce’s disease. Therefore, knowing the diversity and abundance of these insects in Portuguese vineyards is the first step for implementing appropriate measures to control these pathogens. Vineyards distributed in mainland Portugal were sampled to investigate species composition, richness, and diversity of the Cicadomorpha community, focusing on vectors and potential vectors of *X. fastidiosa*. A total of 11,834 individuals belonging to 81 different species/morphospecies were collected. Of these species, only two are confirmed vectors of *X. fastidiosa* (*Philaenus spumarius* and *Neophilaenus campestris*), and three are considered potential vectors (*Cicadella viridis*, *Lepyronia coleoptrata*, and *N. lineatus*). Species that cause direct damage to vines and vectors of grapevine yellows’ phytoplasmas were also collected.

**Abstract:**

Cicadomorpha (Hemiptera) insects are currently responsible for a growing negative impact on the agricultural economy due to their ability to directly damage crops or through the capacity to act as vectors for plant pathogens. The phytopathogenic bacterium *Xylella fastidiosa*, the causal agent of Pierce’s disease in vineyards, is exclusively transmitted by insects of this infraorder. Therefore, knowledge of the Cicadomorpha species and understanding their biology and ecology is crucial. In this work, in 2018 and 2019, the canopy and inter-row vegetation of 35 vineyards distributed in mainland Portugal were sampled to investigate species composition, richness, and diversity of the Cicadomorpha community, with a special focus given to vectors and potential vectors of *X. fastidiosa*. A total of 11,834 individuals were collected, 3003 in 2018 and 8831 in 2019. Of the 81 species/morphospecies identified, only five are considered vectors or potential vectors of this pathogen, namely, *Cicadella viridis* (Linnaeus, 1758), *Philaenus spumarius* (Linnaeus, 1758), *Neophilaenus campestris* (Fallén, 1805), *Lepyronia coleoptrata* (Linnaeus, 1758), and *N. lineatus* (Linnaeus, 1758). *Cicadella viridis* was the most abundant xylem sap feeder, followed by *P. spumarius*. In addition, Cicadomorpha that cause direct damage to vines and vectors of grapevine yellows’ phytoplasmas were also collected and identified in the sampled vineyards. The results suggested that vectors and potential vectors of *X. fastidiosa* and a large proportion of the population of Cicadomorpha have a positive correction with inter-row vegetation.

## 1. Introduction

Viticulture is an important agricultural, environmental, cultural, and economic driving force in the Mediterranean basin. In Portugal, viticulture is an activity of great economic importance, with the vine being cultivated throughout the national territory [1]. Unfortunately, this crop, like others, is subject to pests and diseases that threaten production, quality, and wine typicity [2,3,4,5].

Cicadomorpha (Hemiptera: Auchenorrhyncha) is one of the richest and phyletically diverse infraorders of Hemiptera, with over 30,000 species described worldwide [6]. This infraorder includes exclusively phytophagous species that feed on mesophyll, xylem, or phloem sap [7]. Although most Cicadomorpha do not present a threat to the crops, the infraorder includes several species regarded as economically important pests. These species can damage plants directly through feeding [3,8,9,10] and indirectly through the transmission of plant pathogens [2,4,11,12].

When some individuals of the Cicadomorpha feed, their saliva can induce obstructions on plant vascular tissues, leading to deformations, a discolouration of leaves, or even premature death of plants [13]. In vineyards, the green leafhoppers *Jacobiasca lybica* (Bergevim and Zanon, 1922) and *Empoasca vitis* (Göeth, 1875) (Cicadellidae: Typhlocybinae) are considered key pests due to the direct damage caused when feeding [13,14]. Cicadomorpha is among the most significant groups of vectors of plant pathogens [2,15,16,17]. These hemipterans are considered vectors and potential vectors of two serious plant pathogens in vineyards: i) the causal agent of the Flavescence dorée of the vineyard, the phytoplasma ‘*Candidatus* Phytoplasma vitis’, transmitted exclusively by the Cicadellidae *Scaphoideus titanus* Ball, 1932 (Cicadellidae: Deltocephalinae) [2]; and ii) the xylem-limited bacterium *Xylella fastidiosa* (Wells et al., 1987) (Gammaproteobacteria: Xanthomonadaceae), responsible for Pierce’s disease [18]. This bacterium is transmitted exclusively by xylem sap feeders of the infraorder Cicadomorpha, being the subfamily Cicadellinae (Cicadellidae) and the families Aphrophoridae and Cercopidae, the main groups of potential vectors [12,15]. In Europe, the spittlebugs *Philaenus spumarius* (Linnaeus, 1758), *P. italosigmus* Drosopoulos and Remane (2000), and the *Neophilaenus camprestris* (Fallen, 1805), are confirmed vectors of this pathogen [19].

*Xylella fastidiosa* is a plant endophyte native to the Americas [20,21], which is currently responsible for economic losses in the Californian wine sector of around 92 million euros per year [22]. In Europe, despite a sporadic and unconfirmed report of symptoms of *X. fastidiosa* in vineyards in 1997 [23], the bacterium was declared absent on the continent until 2013 [24]. The first official widespread detection of *X. fastidiosa* was reported in the Lecce Region of Apulia, Italy, where bacteria have already decimated thousands of olive trees [25]. Since this first report, the bacterium has spread to other European countries. Outbreaks have been reported in France, Germany (outbreak eradicated), Spain, and Portugal [26]. In Portugal, the fastidious bacterium was detected in January 2019, in Vila Nova de Gaia, in lavender plants (*Lavandula dentata* Linnaeus) [27]. More recently, new outbreaks were reported in other regions of the country [28]. Since there is no cure for the bacterium, detailed knowledge of the abundance and diversity of potential vectors of *X. fastidiosa* and the remaining adult community of Cicadomorpha in the Portuguese agrosystems is the first step in preventing diseases or minimising its potential effects.

With this in mind, the present work is dedicated to studying the Cicadomorpha community, focusing on the vectors and potential vectors of *X. fastidiosa* in the canopy and in the inter-row vegetation of Portuguese vineyards.

## 2. Materials and Methods

### 2.1. Study Area

The study was conducted for two consecutive years, 2018 and 2019, in 35 vineyards (20 vineyards in both years and an additional 15 in the second year) distributed in mainland Portugal (Appendix A). All vineyards were under sustainable producing systems (integrated or organic), and the inter-rows vegetation was maintained during the sampling periods. Additional information regarding the vineyards’ sampling dates and features can be found in Appendix A. Each vineyard was surveyed in three different periods: late spring, summer, and autumn.

### 2.2. Collection and Identification of Insects

In each vineyard, Cicadomorpha adults were sampled in the inter-row and the canopy of the vines with a standard entomological sweep net (38 cm). In the inter-row of the vineyards, 10 samples of 10 consecutive sweepings randomly distributed over 1 ha were collected. For the 
canopy, ten samples of 50 successive sweepings were collected. The content of 
the sweepings was emptied into a plastic bag properly labelled and sealed. 
Arthropods were sorted under a stereoscopic microscope (Leica Microsystems, Wetzlar, 
Germany) and conserved in 96% ethanol until further identification. All the 
adults of the infraorder Cicadomorpha collected were identified. For species 
identification, the male genitalia was dissected and placed in a heated 
solution of 10% potassium hydroxide (KOH) for between 20 s and 3 min, depending 
on the 3clerotization of each specimen. Subsequently, each genitalia was mounted 
in glass slides with glycerine and observed under a stereoscopic microscope. 
The taxonomic classification was based on appropriate keys and illustrations [11,29,30,31,32]. Females were identified to the lowest 
possible taxonomic level. If all males of a genus in a specific sample belonged 
to one species, then females of that same genus were considered to be that 
species. If there were more than one species in a particular genus, females 
belonging to that genus were identified as morphospecies and designated by 
genus or subfamily, followed by “sp.” and a number according to the morphotype 
(e.g., *Psammotettix* sp.1 or *Deltocephalinae* sp.1).

### 2.3. Data Analysis

The community structure was evaluated in terms of the abundance, richness, and diversity of species/morphospecies. The data for each year of the study, 2018 and 2019, were treated independently to avoid bias from the interannual variability. All the statistical analyses were performed in the R software [33]. The mean and the total number of individuals captured by stratum (inter-row and canopy of the vines) and sampling year were described. The specific richness and two diversity indices (Shannon–Wiener Diversity Index (H’) and Pielou Equitability Index (J’)) were determined using the “vegan” package [34]. The specific richness was calculated as the number of species/morphospecies in each vineyard. The Shannon–Wiener index (H’) is the most used index, and it gives greater importance to rare species [35], while the Pielou Equability Index (J’) is derived from the Shannon diversity index and allows the representation of a uniform distribution of individuals among existing species [36]. To analyze the effect of the sampled stratum in the Cicadomorpha community, a permutational multivariate analysis of variance (PERMANOVA) was performed using the function adonis2 from the package “vegan”. To assess the sampling effort, species accumulation curves were drawn in function of the number of vineyards sampled per stratum. Species accumulation curves were computed using the specaccum function of the “vegan” package. Additionally, a co-inertia analysis (CIA) was performed to determine the relationship between Cicadomorpha species/morphospecies and the year of sampling and stratum. This analysis was performed using the “ade4” package and the table.value function to visualize the results.

## 3. Results

In total, 11,834 individuals were collected, of which 3003 in 2018 and 8831 in 2019 (Table 1). Over the two years of study, 81 species/morphospecies were identified. *Psammotettix* sp.1 (3314 individuals), *E. vitis* (2866 individuals), and *Zyginidia scutellaris* (Herrich-Schäffer, 1838) (1079 individuals) were the most abundant species/morphospecies.

In the canopy of the vines, a total of 4262 individuals were recovered (987 in 2018 and 3275 in 2019). The population was dominated by individuals of the subfamily Typhlocybinae, which represents 92% of the total recovered in the canopy of the vines.

In the inter-row vegetation, 7572 individuals were recovered (2016 in 2018 and 5556 in 2019). The inter-row vegetation was dominated by individuals of the subfamily Deltocephalinae, representing 68% of the total individuals captured in this stratum.

Concerning vectors and potential vectors of *X. fastidiosa*, five species were captured, namely: *C. viridis* (307 individuals), *P. spumarius* (112 individuals), *N. campestris* (62 individuals), *Lepyronia coleoptrata* (7 individuals), and *N. lineatus* (3 individuals); the highest abundance of individuals was observed in the inter-row vegetation in the year 2019.

The specific richness and the Shannon—Wiener Diversity Index (H*’*) were significantly higher in the inter-row vegetation (Table 2). However, the Pielou Equitability Index (J*’*) showed no significant differences between the canopy and inter-row vegetation, indicating a uniform species distribution (Table 2).

According to the NMDS analysis based on the Bray–Curtis index (Figure 1) and the PERMANOVA analysis (df = 1; F = 2; *p* = 0.001 for 2018, and df = 1; F = 5; *p* = 0.001 for 2019), the sampling stratum significantly influences the Cicadomorpha community.

In both years and stratum, the species accumulation curves showed a tendency toward stabilisation (Figure 2), which indicates that the sampling effort was sufficient to detect most of the species of the Cicadomorpha community present in the vineyards.

The majority of the Cicadomorpha species, including vectors and potential vectors of *X. fastidiosa*, showed a positive correlation with inter-row vegetation (Figure 3).

## 4. Discussion

Sustainable agriculture requires knowledge of the abundance and diversity of pests and vector insects to protect crops and implement long-term and safe control measures. This principle is the basis of the present work that allowed the identification of the Cicadomorpha community in Portuguese vineyards together with its preference for the different strata.

All insects that feed exclusively on xylem are considered potential vectors of *X. fastidiosa* [15]. In the European continent, 96 species specialised in xylem have been recorded [37]. Among them, only five were captured in the sampled vineyards, namely, *P. spumarius*, *N. campestris*, *N. lineastus*, *L. coleoptera*, and *C. viridis*. Species such as Aphrophora sp., Cercopis intermedia Kirschbaum 1868, and *Philaenus tesselatus* Melichar, 1899, reported in other Portuguese agroecosystems [38,39,40,41], can be considered potential vectors of this pathogen [15], but were not collected in the sampled vineyards. Until now, only *P. spumarius* and *N. campestris*, were shown to be competent vectors of *X. fastidiosa* [19]. Several studies have demonstrated that *P. spumarius* can efficiently transmit *X. fastidiosa* to vineyards [42,43,44]. Little is known about the other three species’ ability to transmit the bacteria. Nonetheless, according to Bodino et al. [45], when acquiring the pathogen through an artificial diet, *C. viridis* is an inefficient vector of *X. fastidiosa*, since it can transmit the pathogen to periwinkle with very low efficiency but with no successful transmission from plant to plant. Since this insect was the most abundant xylem feeder captured in the sampled Portuguese vineyards, a particular effort should be made to clarify and understand the potential role of the sharpshooter in spreading the fastidious bacterium within this agroecosystem.

*Philaenus spumarius* was the most abundant spittlebug in the sampled vineyards, consistent with other studies carried out in European and Californian vineyards [42,46,47]. However, in the present work, the registered abundance was much lower than those reported in the bibliography.

All the vectors and potential vectors of *X. fastidiosa* collected showed a higher abundance in the inter-row vegetation. In fact, the co-inertia analysis (Figure 3) indicated that all the xylem sap feeders present a positive correlation with the inter-row vegetation, which is in line with the literature [48,49,50,51,52,53]. Spittlebugs and *C. viridis* spend a large part of their life cycle in the vegetation cover, mainly in grasses, where they feed, mate, and lay eggs [15,46,51,54]. Nevertheless, with the exception of *N. lineatus* and *L. coleoptrata*, the remaining xylem feeders recovered were also present in the canopy of the vines. Previous studies have also reported the presence of these insects in the canopy of vines [15,42,46,47]. One factor influencing their distribution between the vine canopy and inter-row vegetation is the hour of the day. It is expected that the movement of the insects from the different strata during the day would occur, but we don’t have observations that corroborate this.

The remaining species of Cicadomorpha captured in the vineyards are phloem or mesophyll feeders. Some studies have reported that some phloem sap feeders of the subfamily Deltocephalinae, the most abundant subfamily in the sampled vineyards, presented to be infected with the bacteria [50,55,56]. However, there is no evidence that they can transmit the pathogen [19,57]. As a result, all the remaining species of Cicadomorpha captured in the sampled vineyards most likely do not threaten the vineyards regarding the transmission of *X. fastidiosa*. Further studies on the ability of these individuals to transmit the bacteria are required.

Nonetheless, it should be noted that in addition to vectors and potential vectors of *X. fastidiosa*, some of the species collected in the sampled vineyards are also considered vectors or potential vectors of yellow disease phytoplasmas responsible for destructive damage in the vineyard. Among these, *S. titanus*, the main vector of the Flavescence dorée phytoplasma [2], should be highlighted. *Euscelidius variegatus* (Kirschbaum, 1858) is another species with potential importance; it demonstrated the ability to acquire and transmit the Flavescence dorée phytoplasma under laboratory conditions [58] and also tested positive for *Candidatus* Phytoplasma solani [59,60]. *Neoaliturus fenestratus* (Herrich-Shaffer, 1834) has been reported to carry the ‘*Candidatus* Phytoplasma solani’ [61,62,63]. *Anaceratagallia glabra Dmitriev,* 2020 (= *A. laevis*), *A. sinuata* (Mulsant and Rey, 1855), and *Z. scutellaris* have also been established as potential vectors of the phytoplasmas of yellow grapevine diseases [61]. It is also noted that the main vector of X. fastidiosa, *P. spumarius*, tested positive for the phytoplasma ‘*Ca*. P. solani’, but there was no evidence of transmission to grapevine [60].

Within the Cicadomorpha community, some species recovered in the sampled vineyards can also cause physical damage to the plants, consequently leading to economic losses. *E. vitis* and *J. lybica*, commonly known as green leafhoppers, are key pests in several European wine-producing regions [3,13]. These green leafhoppers feed by puncturing the phloem vessels of the leaves. This induces an obstruction of the vessels, a reddening, and necrosis of leaves, thus reducing photosynthesis and resulting in delayed maturity [13].

A great abundance, richness, and diversity of Cicadomorpha individuals were observed in the inter-row vegetation over the two years of study. In fact, the co-inertia analysis showed that most of the captured individuals exhibited a positive correlation with the inter-row vegetation, with only 14 species, mostly belonging to the family Typhlocybinae, showing a positive correlation with the vine canopy (Figure 3). Data in agreement with the analysis of PERMANOVA and NMDS showed differences between the communities of the sampled strata. A study by Carpio [49], whose objective was to understand the role of herbaceous vegetation in structuring the Cicadomorpha community, showed that olive groves with herbaceous vegetation showed higher diversity and abundance of Cicadomorpha compared to olive groves without herbaceous vegetation. Other studies also highlight the importance of vegetation cover in structuring the Cicadomoprha community [64,65,66,67,68]. Herbaceous vegetation can provide a wide range of food sources, shelter, mating places, and substrates for laying eggs [69]. Tillage or mowing of the vegetation cover can be one solution to reduce Cicadomorpha population levels in agroecosystems; however, these techniques might have significant side effects. The vegetation cover also provides shelter and food to a wide range of beneficial fauna that performs essential ecosystem services in the vineyard, such as pollination, decomposition, regulation of the nutrient cycle, and control of pests and diseases.

## 5. Conclusions

In conclusion, this study focused on the species composition, richness, and diversity of the Cicadomorpha community in vineyards distributed in mainland Portugal, with special emphasis on vectors and potential vectors of *X. fastidiosa*. The results demonstrate that vectors and potential vectors of this pathogen are present in Portuguese vineyards. *Cicadella viridis* was the more abundant potential vector in the sampled vineyards. Further studies on transmission rates are necessary to better understand this insect’s role in *X. fastidiosa* epidemiology. *Philaenus spumarius*, the main European vector, was the most abundant spittlebug.

Additionally, vectors of phytoplasmas of yellow grapevine diseases and species that can physically damage vines were also collected and identified in the sampled vineyards.

Vectors and potential vectors of *X. fastidiosa* and a large part of the population of Cicadomorpha showed a positive correction with inter-row vegetation.

Further research on how the landscape, agricultural practices, application of phytosanitary treatments, the variety present at the sampling site, and environmental conditions shape the Cicadomorpha community is essential to design new techniques to prevent the spread of this pathogen in Portuguese vineyards.

## Figures and Tables

**Figure 1 insects-14-00251-f001:**
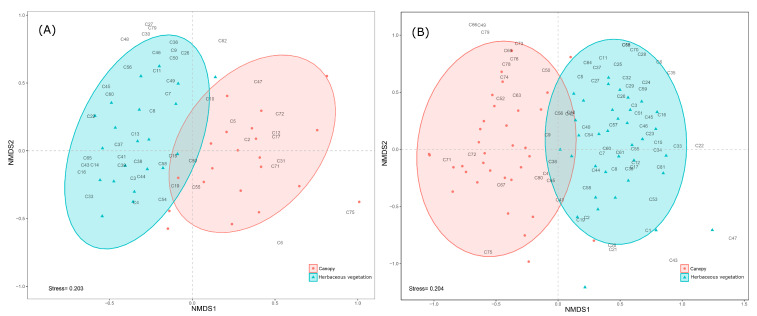
Non-metric multidimensional scaling (NMDS) analysis for Cicadomorpha abundance in the canopy and inter-row vegetation in (**A**) 2018 and (**B**) 2019. The numbers within the panels correspond to the numbers of the species/morphospecies in Table 1.

**Figure 2 insects-14-00251-f002:**
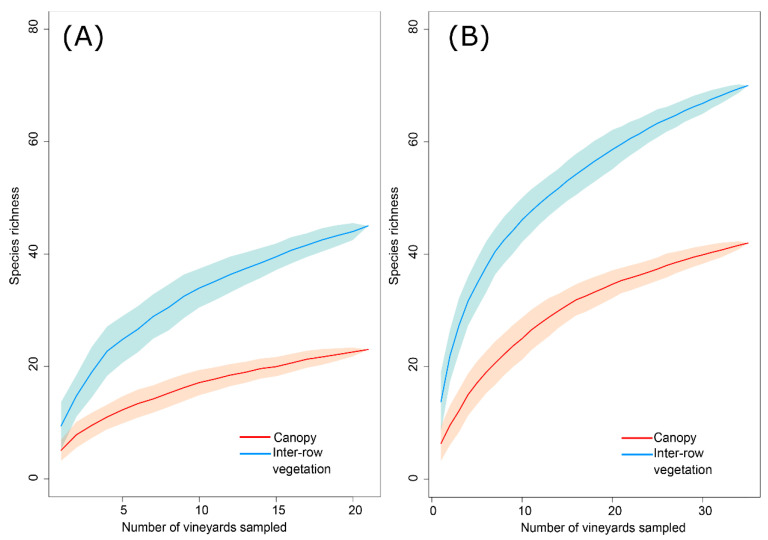
Species accumulation curves based on the amount of vines sampled for (**A**) 2018 and (**B**) 2019 in the canopy and inter-row vegetation. The envelopes correspond to the 95% confidence interval.

**Figure 3 insects-14-00251-f003:**
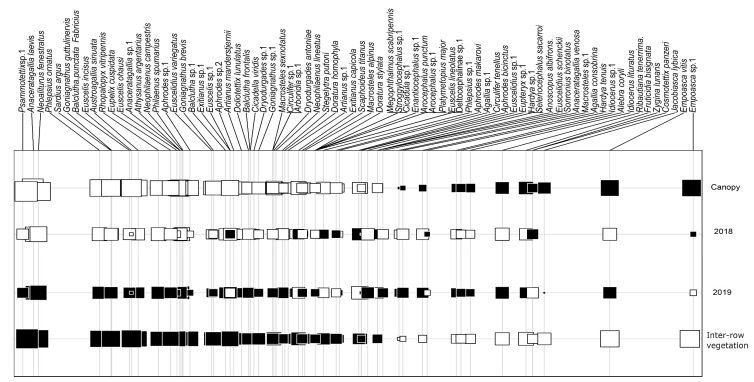
Co-inertia factorial map. Black squares represent positive relationships and white squares negative relationships. Square sizes are proportional to the magnitude of the correlation.

**Table 1 insects-14-00251-t001:** Mean and respective standard error (ME ± SE) and total number (N) of Cicadomorpha adults collected in 2018 and 2019 in the vine canopy and inter-row vegetation.

Family	Subfamily		Species	2018								2019								Total			
				Canopy				Inter-Row Vegetation				Canopy				Inter-Row Vegetation							
				ME	±	SE	N	ME	±	SE	N	ME	±	SE	N	ME	±	SE	N	ME	±	SE	N
Aphrophoridae		C1	*Lepyronia coleoptrata* (Linnaeus. 1758)	0.0	±	0.0	0	0.03	±	0.03	1	0.0	±	0.0	0	0.17	±	0.10	6	0.13	±	0.07	7
		C2	*Neophilaenus campestris* (Fallén. 1805)	0.1	±	0.1	2	0.10	±	0.07	2	0.23	±	0.11	8	1.43	±	0.50	50	1.11	±	0.34	62
		C3	*Neophilaenus lineatus* (Linnaeus. 1758)	0.0	±	0.0	0	0.05	±	0.05	1	0.00	±	0.00	0	0.06	±	0.04	2	0.05	±	0.03	3
		C4	*Philaenus spumarius* (Linnaeus. 1758)	0.2	±	0.1	5	1.29	±	0.34	27	0.46	±	0.16	16	1.83	±	0.67	64	2.00	±	0.45	112
Cicadellidae	Agalliinae	C5	*Agallia consobrina* Curtis. 1833	0.1	±	0.1	3	0.05	±	0.05	1	0.34	±	0.20	12	0.26	±	0.10	9	0.45	±	0.15	25
		C6	*Agallia* sp.1	0.0	±	0.0	1	0.00	±	0.00	0	0.00	±	0.00	0	0.09	±	0.06	3	0.07	±	0.04	4
		C7	*Anaceratagallia glabra* Dmitriev, 2020 (*A. laevis*)	0.1	±	0.1	3	1.57	±	0.80	33	0.46	±	0.17	16	7.51	±	1.67	263	5.63	±	1.21	315
		C8	*Anaceratagallia* sp.1	0.1	±	0.1	2	1.19	±	0.31	25	0.14	±	0.07	5	0.97	±	0.40	34	1.18	±	0.28	66
		C9	*Anaceratagallia venosa* (de Fourcroy. 1785)	0.0	±	0.0	0	0.10	±	0.10	2	0.23	±	0.23	8	0.14	±	0.08	5	0.27	±	0.15	15
		C10	*Austroagallia sinuata* (Mulsant & Rey. 1855)	0.2	±	0.2	5	0.86	±	0.42	18	0.34	±	0.14	12	2.40	±	0.72	84	2.13	±	0.55	119
		C11	*Dryodurgades antoniae* (Melichar. 1907)	0.0	±	0.0	1	1.05	±	0.61	22	0.09	±	0.06	3	0.46	±	0.22	16	0.75	±	0.28	42
		C12	*Dryodurgades* sp.1	0.0	±	0.0	1	0.00	±	0.00	0	0.00	±	0.00	0	0.14	±	0.06	5	0.11	±	0.04	6
	Aphrodinae	C13	*Anoscopus albifrons* (Linnaeus. 1758)	0.0	±	0.0	0	0.05	±	0.05	1	0.00	±	0.00	0	0.00	±	0.00	0	0.02	±	0.02	1
		C14	*Aphrodes bicinctus* (Schrank. 1776)	0.0	±	0.0	0	0.10	±	0.07	2	0.00	±	0.00	0	0.00	±	0.00	0	0.04	±	0.03	2
		C15	*Aphrodes makarovi* Zachvatkin. 1948	0.0	±	0.0	0	0.10	±	0.10	2	0.00	±	0.00	0	0.03	±	0.03	1	0.05	±	0.04	3
		C16	*Aphrodes* sp.1	0.0	±	0.0	0	0.10	±	0.07	2	0.00	±	0.00	0	0.14	±	0.06	5	0.13	±	0.04	7
		C17	*Aphrodes* sp.2	0.0	±	0.0	0	0.00	±	0.00	0	0.00	±	0.00	0	0.46	±	0.25	16	0.29	±	0.16	16
		C18	*Stroggylocephalus* sp.	0.0	±	0.0	0	0.00	±	0.00	0	0.00	±	0.00	0	0.03	±	0.03	1	0.02	±	0.02	1
	Cicadellinae	C19	*Cicadella viridis* (Linnaeus. 1758)	0.2	±	0.1	4	0.43	±	0.18	9	0.06	±	0.04	2	8.34	±	4.95	292	5.48	±	3.12	307
	Deltocephalinae	C20	*Arocephalus punctum* (Flor. 1861)	0.0	±	0.0	0	0.00	±	0.00	0	0.00	±	0.00	0	0.03	±	0.03	1	0.02	±	0.02	1
		C21	*Arocephalus* sp.1	0.0	±	0.0	0	0.00	±	0.00	0	0.00	±	0.00	0	0.03	±	0.03	1	0.02	±	0.02	1
		C22	*Artianus manderstjernii* (Kirschbaum. 1868)	0.0	±	0.0	0	0.05	±	0.05	1	0.00	±	0.00	0	0.51	±	0.30	18	0.34	±	0.19	19
		C23	*Artianus* sp.1	0.0	±	0.0	0	0.00	±	0.00	0	0.00	±	0.00	0	0.11	±	0.09	4	0.07	±	0.06	4
		C24	*Athysanus argentarius* Metcalf. 1955	0.0	±	0.0	0	0.00	±	0.00	0	0.00	±	0.00	0	0.31	±	0.13	11	0.20	±	0.08	11
		C25	*Balclutha frontalis* (Ferrari. 1882)	0.0	±	0.0	0	0.00	±	0.00	0	0.00	±	0.00	0	0.11	±	0.07	4	0.07	±	0.04	4
		C26	*Balclutha punctata* (Fabricius. 1775)	0.0	±	0.0	0	0.48	±	0.27	10	0.00	±	0.00	0	1.57	±	0.53	55	1.16	±	0.35	65
		C27	*Balclutha* sp.1	0.0	±	0.0	0	0.05	±	0.05	1	0.09	±	0.05	3	1.11	±	0.55	39	0.77	±	0.37	43
		C28	*Cicadula* sp.1	0.0	±	0.0	0	0.00	±	0.00	0	0.00	±	0.00	0	0.03	±	0.03	1	0.02	±	0.02	1
		C29	*Circulifer* sp.1	0.0	±	0.0	0	0.00	±	0.00	0	0.00	±	0.00	0	0.06	±	0.04	2	0.04	±	0.03	2
		C30	*Circulifer tenellus* (Baker. 1896)	0.0	±	0.0	0	0.48	±	0.31	10	0.00	±	0.00	0	0.00	±	0.00	0	0.18	±	0.12	10
		C31	*Cosmotettix panzeri* (Flor. 1861)	0.0	±	0.0	1	0.00	±	0.00	0	0.00	±	0.00	0	0.00	±	0.00	0	0.02	±	0.02	1
		C32	*Doliotettix lunulatus* (Zetterstedt. 1838)	0.0	±	0.0	0	0.00	±	0.00	0	0.11	±	0.07	4	4.94	±	2.89	173	3.16	±	1.84	177
		C33	*Doratura homophyla* (Flor. 1861)	0.0	±	0.0	0	0.05	±	0.05	1	0.00	±	0.00	0	0.49	±	0.38	17	0.32	±	0.24	18
		C34	*Doratura stylata* (Boheman. 1847)	0.0	±	0.0	0	0.00	±	0.00	0	0.00	±	0.00	0	0.06	±	0.06	2	0.04	±	0.04	2
		C35	*Enantiocephalus* sp.1	0.0	±	0.0	0	0.00	±	0.00	0	0.00	±	0.00	0	0.03	±	0.03	1	0.02	±	0.02	1
		C36	*Euscelidius schenckii* (Kirschbaum. 1868)	0.0	±	0.0	0	0.05	±	0.05	1	0.00	±	0.00	0	0.00	±	0.00	0	0.02	±	0.38	1
		C37	*Euscelidius variegatus* (Kirschbaum. 1858)	0.0	±	0.0	0	0.38	±	0.10	8	0.06	±	0.04	2	1.14	±	0.57	40	0.89	±	0.02	50
		C38	*Euscelidius* sp.1	0.0	±	0.0	0	0.10	±	0.07	2	0.06	±	0.06	2	0.06	±	0.06	2	0.11	±	0.91	6
		C39	*Euscelis incisus* (Kirschbaum. 1858)	0.0	±	0.0	1	3.29	±	1.76	69	0.03	±	0.03	1	3.57	±	0.99	125	3.50	±	0.04	196
		C40	*Euscelis lineolatus* Brullé. 1832	0.0	±	0.0	0	0.00	±	0.00	0	0.00	±	0.00	0	0.06	±	0.06	2	0.04	±	0.08	2
		C41	*Euscelis ohausi* W.Wagner. 1939	0.0	±	0.0	0	0.00	±	0.00	0	0.03	±	0.03	1	0.23	±	0.11	8	0.16	±	0.61	9
		C42	*Euscelis* sp.1	0.0	±	0.0	1	1.57	±	0.82	33	0.20	±	0.09	7	2.09	±	0.80	73	2.04	±	4.42	114
		C43	*Exitianus capicola* (Stål. 1855)	0.9	±	0.7	18	17.48	±	9.46	367	0.06	±	0.06	2	9.46	±	3.70	331	12.82	±	0.41	718
		C44	*Exitianus* sp.1	0.0	±	0.0	0	1.62	±	0.99	34	0.00	±	0.00	0	0.40	±	0.26	14	0.86	±	0.21	48
		C45	*Goniagnathus brevis* (Herrich-Schäffer. 1835)	0.0	±	0.0	0	0.29	±	0.17	6	0.00	±	0.00	0	0.63	±	0.32	22	0.50	±	0.45	28
		C46	*Goniagnathus guttulinervis* (Kirschbaum. 1868)	0.0	±	0.0	0	0.57	±	0.31	12	0.03	±	0.03	1	2.17	±	0.68	76	1.59	±	0.10	89
		C47	*Goniagnathus* sp.1	0.0	±	0.0	1	0.05	±	0.05	1	0.00	±	0.00	0	0.29	±	0.16	10	0.21	±	0.02	12
		C48	*Hardya* sp.1	0.0	±	0.0	0	0.05	±	0.05	1	0.00	±	0.00	0	0.00	±	0.00	0	0.02	±	0.03	1
		C49	*Hardya tenuis* (Germar. 1821)	0.0	±	0.0	0	0.05	±	0.05	1	0.03	±	0.03	1	0.00	±	0.00	0	0.04	±	0.07	2
		C50	*Macrosteles alpinus* (Zetterstedt. 1828)	0.0	±	0.0	0	0.10	±	0.10	2	0.03	±	0.03	1	0.11	±	0.09	4	0.13	±	0.08	7
		C51	*Macrosteles sexnotatus* (Fallén. 1806)	0.0	±	0.0	0	0.00	±	0.00	0	0.00	±	0.00	0	0.17	±	0.12	6	0.11	±	0.05	6
		C52	*Macrosteles* sp.1	0.0	±	0.0	0	0.00	±	0.00	0	0.09	±	0.06	3	0.06	±	0.06	2	0.09	±	0.06	5
		C53	Deltocephalinae sp.1	0.0	±	0.0	0	0.00	±	0.00	0	0.00	±	0.00	0	0.77	±	0.77	27	0.48	±	0.48	27
		C54	*Neoaliturus fenestratus* (Herrich-Schäffer. 1834)	0.2	±	0.1	5	0.62	±	0.23	13	0.37	±	0.14	13	3.34	±	0.70	117	2.64	±	0.52	148
		C55	*Phlepsius ornatus* (Perris. 1857)	0.0	±	0.0	1	0.10	±	0.07	2	0.00	±	0.00	0	0.71	±	0.20	25	0.50	±	0.13	28
		C56	*Phlepsius* sp.1	0.0	±	0.0	0	0.24	±	0.14	5	0.03	±	0.03	1	0.06	±	0.06	2	0.14	±	0.06	8
		C57	*Platymetopius major* (Kirschbaum. 1868)	0.0	±	0.0	0	0.00	±	0.00	0	0.00	±	0.00	0	0.06	±	0.06	2	0.04	±	0.04	2
		C58	*Psammotettix* sp.1	1.9	±	0.5	39	37.81	±	11.44	794	3.09	±	0.76	108	67.80	±	15.43	2373	59.18	±	10.79	3314
		C59	*Rhopalopyx vitripennis* (Flor. 1861)	0.0	±	0.0	0	0.00	±	0.00	0	0.00	±	0.00	0	0.63	±	0.23	22	0.39	±	0.15	22
		C60	*Sardius argus* (Marshall. 1866)	0.0	±	0.0	0	0.38	±	0.13	8	0.09	±	0.05	3	1.26	±	0.39	44	0.98	±	0.25	55
		C61	*Scaphoideus titanus* Ball. 1932	0.0	±	0.0	0	0.00	±	0.00	0	0.09	±	0.06	3	2.34	±	2.03	82	1.52	±	1.27	85
		C62	*Selenocephalus sacarroi* Rodrigues. 1968	0.0	±	0.0	0	0.05	±	0.05	1	0.00	±	0.00	0	0.00	±	0.00	0	0.02	±	0.02	1
		C63	*Sonronius binotatus* (Sahlberg. 1871)	0.0	±	0.0	0	0.00	±	0.00	0	0.03	±	0.03	1	0.03	±	0.03	1	0.04	±	0.04	2
		C64	*Stegelytra putoni* Mulsant & Rey. 1875	0.0	±	0.0	0	0.00	±	0.00	0	0.03	±	0.03	1	0.11	±	0.06	4	0.09	±	0.05	5
		C65	*Eupelix cuspidata* (Fabricius. 1775)	0.0	±	0.0	0	0.19	±	0.09	4	0.09	±	0.05	3	0.57	±	0.21	20	0.48	±	0.06	27
	Idiocerinae	C66	*Idiocerus lituratus* (Fallén. 1806)	0.0	±	0.0	0	0.00	±	0.00	0	0.03	±	0.03	1	0.00	±	0.00	0	0.02	±	0.02	1
		C67	*Idiocerus* sp.1	0.0	±	0.0	0	0.00	±	0.00	0	0.14	±	0.12	5	0.03	±	0.03	1	0.11	±	0.05	6
	Megophthalminae																						
		C68	*Megophthalmus scabripennis* Edwards. 1915	0.0	±	0.0	0	0.00	±	0.00	0	0.00	±	0.00	0	0.03	±	0.03	1	0.02	±	0.02	1
	Typhlocybinae	C69	*Alebra coryli* Le Quesne. 1977	0.0	±	0.0	0	0.00	±	0.00	0	0.63	±	0.63	22	0.06	±	0.06	2	0.43	±	0.39	24
		C70	*Arboridia* sp.1	0.0	±	0.0	0	0.00	±	0.00	0	0.00	±	0.00	0	0.06	±	0.04	2	0.04	±	0.03	2
		C71	*Empoasca* sp.1	8.2	±	3.9	172	2.05	±	1.25	43	7.83	±	1.68	274	0.86	±	0.33	30	9.27	±	1.82	519
		C72	*Empoasca vitis* (Göthe. 1875)	18.4	±	6.1	386	6.29	±	2.50	132	59.97	±	2099	17.18	7.11	±	1.94	249	51.18	±	11.87	2866
		C73	*Eupteryx* sp.1	0.0	±	0.0	0	0.00	±	0.00	0	0.14	±	0.07	5	0.14	±	0.12	5	0.18	±	0.10	10
		C74	*Fruticidia bisignata* (Mulsant & Rey. 1855)	0.0	±	0.0	0	0.00	±	0.00	0	0.11	±	0.05	4	0.00	±	0.00	0	0.07	±	0.03	4
		C75	*Jacobiasca lybica* (de Bergevin & Zanon. 1922)	13.8	±	7.4	289	0.00	±	0.00	0	13.29	±	12.12	465	0.26	±	0.19	9	13.63	±	8.06	763
		C76	*Ribautiana tenerrima* (Herrich-Schäffer. 1834)	0.0	±	0.0	0	0.00	±	0.00	0	0.46	±	0.21	16	0.06	±	0.06	2	0.32	±	0.15	18
		C77	*Zygina lunaris* (Mulsant & Rey. 1855)	0.0	±	0.0	1	0.00	±	0.00	0	0.00	±	0.00	0	0.00	±	0.00	0	0.02	±	0.02	1
		C78	*Zygina ordinaria* (Ribaut. 1936)	0.0	±	0.0	0	0.00	±	0.00	0	0.14	±	0.06	5	0.03	±	0.03	1	0.11	±	0.05	6
		C79	*Zygina* sp.1	0.0	±	0.0	0	0.10	±	0.10	2	1.43	±	1.40	50	0.00	±	0.00	0	0.93	±	0.87	52
		C80	*Zyginidia scutellaris* (Herrich-Schäffer. 1838)	2.6	±	0.6	55	14.52	±	5.13	305	2.43	±	0.47	85	18.11	±	3.75	634	19.27	±	3.11	1079
	Ulopinae	C81	*Uteca* sp.1	0.0	±	0.0	0	0.00	±	0.00	0	0.00	±	0.00	0	0.03	±	0.03	1	0.02	±	0.02	1
Total				12.01	±	6.12	987	24.30	±	11.11	2016	39.46	±	25.97	2099	66.94	±	29.92	5556	146.10	±	65.13	11834

**Table 2 insects-14-00251-t002:** Cicadomorpha richness and diversity indices for each stratum sampled per year. Mean Richness, Shannon—Wiener Diversity Index (H’), and Pielou Equitability Index (J’).

		2018			2019		
Richness	Canopy	4.95	±	0.5	6.43	±	0.68
	Inter-row vegetation	9.62	±	0.9	13.86	±	0.10
	*p-*value	<0.001			<0.001		
H’	Canopy	1.00	±	0.10	1.01	±	0.10
	Inter-row vegetation	1.44	±	0.09	1.75	±	0.11
	*p-*value	0.002			<0.001		
J’	Canopy	1.48	±	0.60	1.42	±	0.41
	Inter-row vegetation	1.55	±	0.40	1.60	±	0.43
	*p-*value	0.64			0.14		

## Data Availability

Data are available from the authors upon reasonable request.

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
