# Peer review of "Cicadomorpha Community (Hemiptera: Auchenorrhyncha) in Portuguese Vineyards with Notes of Potential Vectors of Xylella fastidiosa"

_insects, 2023, doi:10.3390/insects14030251_

Round 1

Reviewer 1 Report

The submitted manuscript, "Cicadomorpha community (Hemiptera: Auchenorrhyncha) in Portuguese vineyards with notes of potential vectors of Xylella fastidiosa," by Isabel Rodrigues, Maria Teresa Rebelo, Paula Baptista, and José Alberto Pereira, is a thorough study that is usually missing as a basic study for understanding and predicting vector-borne diseases in agro-ecosystems. Over the course of two years, the authors collected 11834 individuals from more than 80 morphospecies. The numbers are impressive. The collected diversity of Auchenorrhyncha is a solid basis for any further study of potential risks in the emergence of Xylella fastidios as an aggressive plant pathogen. This type of study appeals to me because it allows us to track changes within the Auchenorrhyncha community before an outbreak occurs. Two species, namely, Philaenus spumarius and Neophilaenus campestris, are listed as potential vectors, and their abundance in the collected crop was significant, which is of concern in the vineyard ecosystems and should be carefully followed.

The methods applied in this study are relevant and give the opportunity to compare the collected material with additional future results. Material collected with a sweeping net, in particular, provides an opportunity to study other plant pathogens transmitted by Auchenorrhyncha. The applied statistical methods fit the purpose of such a study. The results and discussion are well presented and satisfactory.

Further research on the landscape and agricultural practices, as stated by the authors, is required to shape the Cicadomorpha community in order to design and predict the spread of plant pathogens transmitted by these insects.
I recommend that this article be published in Insect.

Author Response

We sincerely appreciate all comments and encouraging words given by the Reviewer.

Reviewer 2 Report

Comments and Suggestions for Authors: 

This work represents an important contribution to the knowledge about the Cicadomorpha community in Portuguese vineyards.

I only have a few suggestions:

Page 1

line 16-17: Change Pierce disease for Pierce's disease

line 29: Change Pierce disease for Pierce's disease

Page 2

line 61: delete (Cicadellidae: Typhlocybinae)

line 64: Change Cicadomorpha for These insects ( or These hemipterans)

line 67: (Cicadellidae: Deltocephalinae) after Ball, 1932

line 73-74: This work refers to species of Cicadomorpha, therefore, it is redundant to write "Cicadomorpha" to each species.

Page 3

line 105-106: use 10 instead of ten

line 144-145: species names in italics and if the genus was not previously named it should not be abbreviated

Page 4

line 153-155: species names in italics and if the genus was not previously named it should not be abbreviated

Page 8

C68: subfamily does not match species row

Page 10

line 186: X. fastidiosa in italics

Page 12

Lines 199 to 203, 205 to 208, 210, 216, 220, 223, 225, 237, 239, 240, 242 ans 243: please, put specific names in italics

Page 13

Lines 246 to 251 and 255: please, specific names in italics

Author Response

This work represents an important contribution to the knowledge about the Cicadomorpha community in Portuguese vineyards.

 Author's response: We sincerely appreciate all valuable comments and suggestions given by the Reviewer. which made it possible to improve the manuscript.

I only have a few suggestions:

Page 1

line 16-17: Change Pierce disease for Pierce's disease

Author's response: changed as suggested.

line 29: Change Pierce disease for Pierce's disease

Author's response: changed as suggested.

Page 2

line 61: delete (Cicadellidae: Typhlocybinae)

Author's response: changed as suggested.

line 64: Change Cicadomorpha for These insects ( or These hemipterans)

Author's response: changed as suggested.

line 67: (Cicadellidae: Deltocephalinae) after Ball, 1932

Author's response: changed as suggested.

line 73-74: This work refers to species of Cicadomorpha, therefore, it is redundant to write "Cicadomorpha" to each species.

Author's response: changed as suggested.

Page 3

line 105-106: use 10 instead of ten

Author's response: changed as suggested.

line 144-145: species names in italics and if the genus was not previously named it should not be abbreviated

Author's response: changed as suggested.

Page 4

line 153-155: species names in italics and if the genus was not previously named it should not be abbreviated

Author's response: changed as suggested.

Page 8

C68: subfamily does not match species row

Author's response: the table has been corrected. Please see table 1

Page 10

line 186: X. fastidiosa in italics

Author's response: changed as suggested.

Page 12

Lines 199 to 203, 205 to 208, 210, 216, 220, 223, 225, 237, 239, 240, 242 ans 243: please, put specific names in italics

Author's response: changed as suggested.

Page 13

Lines 246 to 251 and 255: please, specific names in italics

Author's response: changed as suggested.

Reviewer 3 Report

Reviewer: 1

Ms. Ref. No.: Insects-2248560

Authors: Isabel Rodrigues, Maria Teresa Rebelo, Paula Baptista and José Alberto Pereira.

Specific notes:

TITLE

Nothing to comment

SIMPLE SUMMARY

Nothing to comment

ABSTRACT

Line 40: I suggest authors change the word "Our" by the word "The".

KEYWORDS

Nothing to comment

INTRODUCTION

Line 76: “Xylella fastidiosa” should be appear abbreviated. Its full scientific name appears in the previous paragraph (see line 68).

MATERIAL AND METHODS

Line 133: Insert a blank space before the word "To".

RESULTS

Lines 144-145: Psammotettix, E.vitis and Zyginidia scutellaris should appear in italics

Lines 153-157: All cientific names should appear in italics

Line 186: X. fastidiosa should appear in italics

DISCUSSION

Lines 194-255: All cientific names should appear in italics

CONCLUSIONS

Line 283: Cicadella viridis should appear abreviated.

Line 285: Philaenus spumarius should appear abreviated.

FIGURES

Nothing to comment

TABLES

Nothing to comment

REFERENCES

Line 360: X. fastidiosa should appear in italics

Line 365: X. fastidiosa should appear in italics

Line 374: X. fastidiosa should appear in italics

Lines 412-413: ¿Xylella Fastidiosa?

Line 419: X. fastidiosa should appear in italics

Author Response

Authors: Isabel Rodrigues, Maria Teresa Rebelo, Paula Baptista and José Alberto Pereira.

Specific notes:

TITLE

Nothing to comment

SIMPLE SUMMARY

Nothing to comment

ABSTRACT

Line 40: I suggest authors change the word "Our" by the word "The".

Author's response: changed as suggested.

KEYWORDS

Nothing to comment

INTRODUCTION

Line 76: “Xylella fastidiosa” should be appear abbreviated. Its full scientific name appears in the previous paragraph (see line 68).

Author's response: In terms of style, we prefer to keep the complete name of the genus at the beginning of a sentence or paragraph.

MATERIAL AND METHODS

Line 133: Insert a blank space before the word "To".

Author's response: changed as suggested.

RESULTS

Lines 144-145: Psammotettix, E.vitis and Zyginidia scutellaris should appear in italics

Author's response: changed as suggested.

Lines 153-157: All cientific names should appear in italics

Author's response: changed as suggested.

Line 186: X. fastidiosa should appear in italics

Author's response: changed as suggested.

DISCUSSION

Lines 194-255: All cientific names should appear in italics

Author's response: changed as suggested.

CONCLUSIONS

Line 283: Cicadella viridis should appear abreviated.

Line 285: Philaenus spumarius should appear abreviated.

  Author's response: As mentioned, we prefer to keep the complete name of the genus at the beginning of a sentence or paragraph.

Line 360: X. fastidiosa should appear in italics

Author's response: changed as suggested.

Line 365: X. fastidiosa should appear in italics

Author's response: changed as suggested.

Line 374: X. fastidiosa should appear in italics

Author's response: changed as suggested.

Lines 412-413: ¿Xylella Fastidiosa?

Corrected

Line 419: X. fastidiosa should appear in italics

Author's response: changed as suggested.